# Monitoring of Subsurface Emissions and the Influence of Meteorological Factors on Landfill Gas Emissions: A Case Study of a South African Landfill

**Prince Obinna Njoku** [1,*], **Stuart Piketh** [2], **Rachel Makungo** [3] **and Joshua Nosa Edokpayi** [3]

1 Department of Geography and Environmental Sciences, Faculty of Science, Engineering and Agriculture, University of Venda, Thohoyandou 0950, South Africa
2 Climatology Research Group, Unit for Environmental Sciences and Management, North-West University, Potchefstroom 2531, South Africa
3 Department of Earth Sciences, Faculty of Science, Engineering and Agriculture, University of Venda, Thohoyandou 0950, South Africa
* Correspondence: pnjoku26@yahoo.com

**Abstract:** The government-accredited GA 2000 landfill gas analyzer was used to monitor the landfill gas (methane ($CH_4$) and carbon dioxide ($CO_2$)) generated from the subsurface of a landfill. Eighteen gas sample probes were constructed and placed approximately 100 m apart on the boundaries of the landfill site. The monitoring of the gases was conducted over a period of two years, taking into consideration the different seasons of the year. Results from the study show that as the LFG migrates toward the boundaries of the landfill in the subsurface, higher $CO_2$ levels are recorded when compared to $CH_4$. This could be a result of the oxidation process and some other factors present in the landfill. $CH_4$ emissions ranged from 0.54 to 2.22% $v/v$ in 2020. For the year 2021, the $CH_4$ concentration ranged from 0.24% $v/v$ to 2.33% $v/v$. In addition, the $CO_2$ concentration for the year 2020 ranged from 4.66 to 6.37% $v/v$. In the year 2021, the $CO_2$ concentration ranged from 3.5 to 6.56% $v/v$. Furthermore, higher gaseous concentrations were found in the surrounding active cells, where there is currently deposition of municipal solid waste (MSW). However, the monitoring probes situated in areas far away from the landfill recorded lower gaseous levels. This study recommends that there should be continuous monitoring of the LFG emission from the Thohoyandou landfill due to the different results obtained over the temporal variation. Frequent monitoring will help prevent the potential risk of fire hazards and pollution in the future.

**Keywords:** GA 2000 landfill gas analyzer; landfill gas; meteorological data; subsurface emissions

## 1. Introduction

Landfills are one of the main contributors to the world's anthropogenic greenhouse gas (GHG) emissions because massive amounts of methane ($CH_4$) and carbon dioxide ($CO_2$) are emitted from the degradation process of deposited waste in them [1]. In landfills, LFG is produced in three stages: bacterial degradation, chemical reactions, and volatilization. Due to complex physical, chemical, and microbiological processes, persistent organic pollutants (such as dioxins and polycyclic aromatic hydrocarbons), $CH_4$, $CO_2$, heavy metals, non-metallic organic compounds (NMOCs), particulate matter, and some trace elements are regularly generated in landfills [2,3]. In addition, landfill activities generate leachates; these leachates are mostly produced from rainfall, surface water from the surrounding environment, and the decomposition of waste buried in the landfill. The leachate filters through the waste and leaches or draws out several chemical substances from the waste piles. Subsequently, the leachate produced during this process is a large contributor to the rise in odor levels, contaminated groundwater, and breeding grounds for insects [4]. Sharma (2020) [5] identified that leachate generated from an open dumpsite in Himachal

Pradesh, India, was responsible for most of the physicochemical parameters and heavy metals of the groundwater being in excess of the permissible limits and that the water quality improved with an increase in downstream distance from the dumpsite. Many toxic compounds generated by landfill operations have caused concerns among people who live near landfill sites [6]. Other concerns associated with waste deposition in landfills include litter, dust, rodents, and unexpected landfill fires. LFG subsurface migrations can cause severe landfill fires if not properly monitored and managed [1]. Landfill gas subsurface migration is the process by which LFG moves through the subsurface environment. The gases produced migrate through the soil and groundwater and can eventually reach the surface. This process can be accelerated by the presence of fractures or other pathways in the subsurface [7].

*LFG Monitoring*

Monitoring of gases emitted by landfills is divided into five categories: soil gas monitoring (subsurface gas monitoring); near-surface gas monitoring; ambient air monitoring; indoor air monitoring [8].

- Soil gas monitoring:

Soil gas monitoring measures the concentrations of gases in the pore space of soils. Measurements of soil gas levels are taken at the depth of the landfill with the use of probes or wells. Probes or wells remove the flammable $CH_4$ component of LFG as it is generated, allowing it to be flared or used as fuel. Scholars have suggested that the handheld GA 2000 landfill gas analyzer is the preferred instrument for government organizations monitoring landfill gas emissions. Nevertheless, other experimental instruments have been developed as an alternative to the government-approved device for measuring subsurface landfill gas [9,10].

- Near-surface monitoring:

Near-surface monitoring is the measurement (usually by portable instruments) of gas concentrations within a few inches of the surface of the landfill. The monitoring of LFG close to the surface is executed to determine the need for, and the design of, an LFG control system. The near-surface monitoring is also used to determine if an LFG control system is adequately preventing $CH_4$ and other LFGs from escaping in high quantities through the landfill cover. Wang-Yao et al. [11] conducted a study on the seasonal variation in LFG emissions from seven landfills in Thailand. The study used a static chamber technique to measure the LFG fluxes from September to November 2005 (dry season) and January to February 2006 (wet season). The results showed that the spatial variability of LFG emissions in the wet season was higher than that in the dry season, ranging from 0 to 825.79 $g/m^2/day$ and from 0 to 686.93 $g/m^2/day$, respectively. The authors concluded that the higher moisture content present in the landfill during the wet season was responsible for the higher LFG emissions. Other studies conducted on near-surface monitoring include the works of Scheutz et al. [12], Park and Shin [13], and Fredenslund et al. [14].

- Atmospheric monitoring

Atmospheric monitoring assesses the amount of contamination in the air that is breathed by individuals or in the open air. Monitoring of the ambient air at or near landfills is primarily conducted to assess the worker and community exposure risks related to airborne discharges of harmful substances [15]. Studies conducted around the world have demonstrated that long-term exposure to $PM_{2.5}$, $SO_2$, NO, $NO_2$, and other air pollutants can have a detrimental effect on health, leading to an increased risk of disease and even premature death [15,16].

During the processes of the breaking down of municipal solid waste (MSW) in landfills, the major components of LFG generated are $CH_4$ and $CO_2$, which comprise approximately 90% of the total LFG generated [17]. $CH_4$ generated from landfills is extremely flammable and poses a significant potential risk of fire outbreak in landfills and the surrounding area. When the volume or concentration of $CH_4$ is in the range of 5–15% and is exposed

to the air at a temperature range of 15–45 °C, it becomes very explosive [18]. According to DAWF (1998), the concentration of $CH_4$ in the atmosphere inside or near any South African landfill site shall not exceed 1% (by volume) in air, or 20% of the lower explosion limit (LEL). Regular monitoring is required if methane levels in the air are discovered to be between 0.1% and 1% (i.e., between 2% and 20% of the LEL). If levels of more than 1% (i.e., 20% of the LEL) are found, the building must be evacuated and trained personnel contacted. The amount of $CH_4$ in the air near landfill boundaries should not exceed 5%, i.e., 100% of the LEL [19]. As a result of regulatory enforcement, landfill emission levels, including subsurface gas migration, must be regularly monitored. Monitoring of these gases from the top of perimeter drilling monitoring pipes and using a hand-held landfill gas analyzer or other techniques is required once a month on average and can be as seldom as four times per year in other instances [18,20]. According to the Environmental Protection Agency, Ireland, the monitoring of the perimeter borehole wells is required once a month. An incident report must be filed with the Office of Environmental Enforcement (OEE) if the $CH_4$ and $CO_2$ concentrations surpass specific criteria, such as 1.0% $v/v$ for $CH_4$ and 1.5% $v/v$ for $CO_2$ [21]. However, Kiernan et al. [21] argue that monitoring the subsurface movements of gases once per month is inadequate to provide a realistic description of the flow and buildup of gases in landfills. This is because the dynamics of the landfill gas management system and activities cannot be captured by taking measurements only once per month; thus, a minimum sampling rate of once per day is advised. Notwithstanding, the monitoring of LFG once per month cannot be entirely flawed because landfill-disturbing activities do not happen all the time but sparingly. This brings LFG flows to normal if there are no forms of disturbances in the landfill. Knowing the above, this study hypothesizes that the Thohoyandou landfill site generates LFG that flows in the subsurface around the boundaries of the landfill. This study seeks to monitor the subsurface emissions of gases ($CH_4$ and $CO_2$) from the Thohoyandou landfill using the GA 2000 landfill gas analyzer.

Several landfill fire outbreaks have occurred in South Africa, such as the incident on the 10th of August 2016, when a section of the New England Road landfill in Pietermaritzburg erupted in flames, emitting a foul odor. At the Woodstock Road informal colony, roughly 70 houses were burned down, leaving at least 140 people homeless. The predominant LFG generated by the waste piles, as well as the dry weather, was responsible for the incident [22]. Therefore, because there are no forms of proper management and control of LFG generated from the rural landfills, there is a high tendency of fire outbreaks in the landfills and their vicinity. The Thohoyandou landfill has suffered from a series of fire outbreaks over the years (key informant: landfill manager). Therefore, it is important to monitor LFG emissions to prevent potential pollution and the risk of landfill fire.

Wastes deposited in landfills are usually compacted in horizontal strata, creating low-permeability barriers and poor vertical LFG flow [23]. Landfills are compacted daily with high-permeability top layers to prevent water intrusion, odor, and gaseous emissions. As a result, rather than the LFG escaping vertically through the landfill's surface, LFG will frequently travel horizontally toward the landfill's boundaries, where it will be discharged into the surroundings through sinkholes and cracks, posing a threat of fire explosion [24]. LFG migration in unsaturated soils around the landfill is mostly influenced by soil physical characteristics such as water content and soil permeability, particularly in the deeper layers of the soil strata. Microbial activity, air pressure, wind speed, temperature, nutrient availability, and oxygen content at the soil surface also influence gas movement and composition [5,25]. In addition, meteorological parameters such as barometric pressure, wind speed, rainfall, and temperature do influence the emissions of LFG. In order to understand the flow rate and travel of the LFG, it is crucial to comprehend how these parameters affect LFG emissions [26,27]. Studies have shown that changes in barometric pressure have a major influence on LFG migration. Furthermore, the effects of wind turbulence-induced pressure variations on soil-gas migration cause a significant fluctuation in gas movement in the upper layer of the soil [28,29]. Likewise, it has been observed that soil water content enhances microbial activities, thereby increasing microbial methane oxidation rates and

having a significant impact on LFG migration in the soil [28,30]. Therefore, the specific objective of this study is to investigate the influence of the meteorological conditions of the Thohoyandou area on LFG generation. This study hypothesizes that meteorological factors around the Thohoyandou environs influence the LFG generation. This study also seeks to investigate the effects of meteorological conditions in the Thohoyandou area on the LFG generated and the lateral movement. The next section gives a brief detailed report on the study area and the comprehensive methods used in carrying out the study.

## 2. Methodology

### 2.1. Study Area

The Thohoyandou landfill is located in Thulamela Municipality in Muledani and manages the MSW from Thohoyandou town and some villages in the Thulamela Municipal region (Figure 1). Thulamela Municipality is 180 km south of Polokwane and serves as a gateway to the Kruger National Park. It is located within the geographical coordinates of 23°0013.0 S and 30°2755.3 E. The Municipality has a population of 584,257 and is largely made up of rural communities. Every year, Thohoyandou receives about 752 mm of rain, with most of its rain falling in the summer (December, January, and February) [31]. The landfill was granted its permit to commence operation in 2004 by the then Minister of Water Affairs and Forestry; i.e., the landfill has been operational for 19 years [32]. The landfill has around four cells that have been closed and one cell that is currently receiving waste. The cell currently receiving waste will soon reach its full lifespan and will be closed permanently. A new cell has been recently constructed and will be operational when the old functioning cell closes. According to Njoku et al. [31], the Thohoyandou landfill's annual composition of waste sources is as follows: commercial and industrial waste (7%), construction and demolition waste (3%), municipal waste (72%), garden waste (7%), and inert waste (10%).

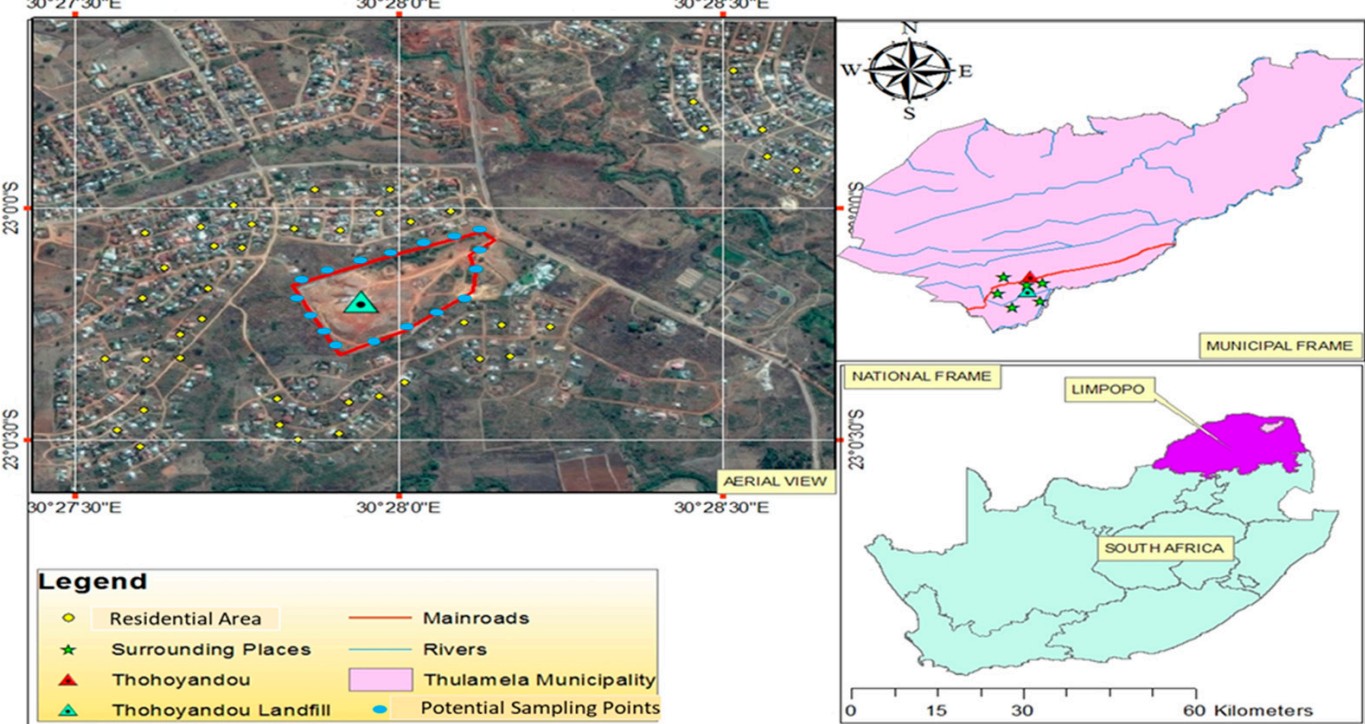

**Figure 1.** Study area map.

### 2.2. Design and Installation of Monitoring Probes

A reconnaissance survey was conducted before the LFG monitoring probes were installed in the landfill. A proper assessment was conducted to ascertain where the LFG monitoring probes would be located. Critical areas between the landfill and adjacent buildings such as groves of trees, utility lines, and fracture zones were assessed before installation. After a proper assessment was conducted, the LFG monitoring probes were installed around the perimeter of the landfill site. The landfill boundary areas were not far from the disposal site.

The landfill gas monitoring probes were designed with expert precise measurements for easy monitoring of the subsurface migration of the LFG using the method of [33]. This was to be able to monitor a good representation of the lateral movement of the gases. A proper LFG monitoring probe was designed to minimize air intrusion into the system so accurate gas sample measurements could be collected from the probes. If air enters the probes, it can dilute the samples, making them unrepresentative.

The LFG monitoring probes were made from PVC pipes; we were careful not to use metals to avoid vandalization or theft. These pipes were perforated along the sides to allow the ingress of LFG into the probes (Figure 2). According to Agency for Toxic Substances and Disease Registry (ATSDR), LFG levels near landfill boundaries must be monitored in MSW landfills. If LFG concentrations surpass the LEL, the smallest percentage by volume of an explosive gas in the atmosphere, there is the risk of an explosion at the monitoring stations at the site boundaries [8]. Bhailall et al. [33], during the monitoring of LFG at a South African landfill, installed the monitoring probes at the boundaries of the landfill site. This was for effective monitoring and to avoid major disturbances in the landfill area. The probes were installed in November 2019. Eighteen monitoring probes were installed at the boundaries of the landfill at a depth of approximately 1–3 m. Figure 3 shows the sampling points where the monitoring probes were installed. The probes were installed within the range of 60 to 150 m apart.

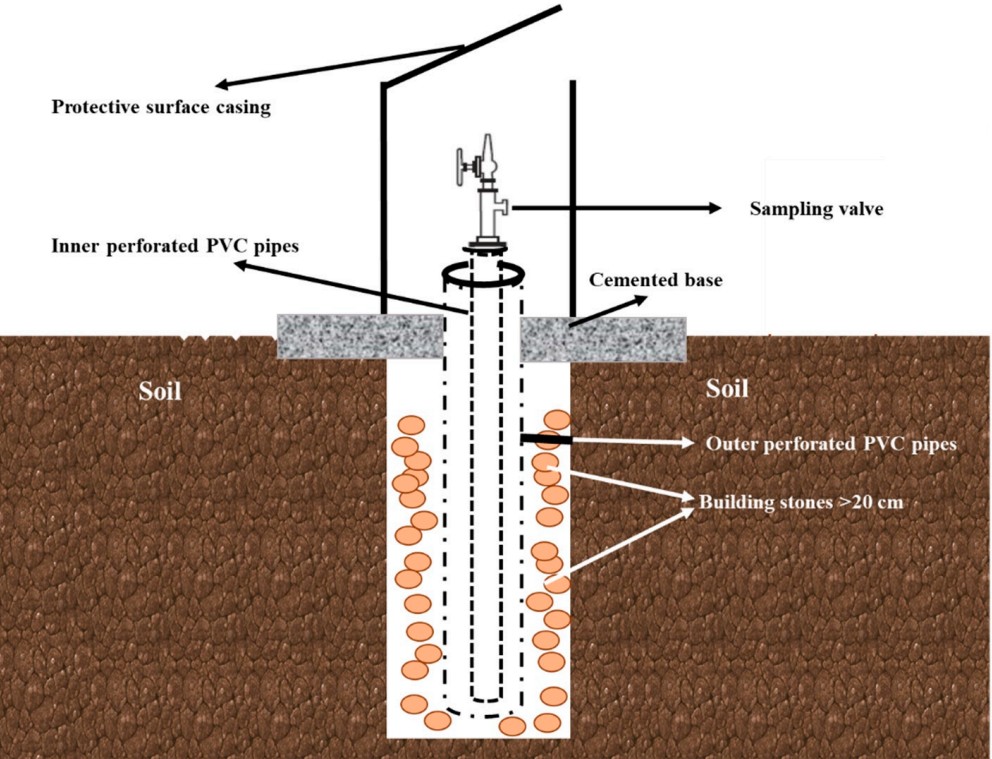

**Figure 2.** A schematic diagram of an installed LFG monitoring probe.

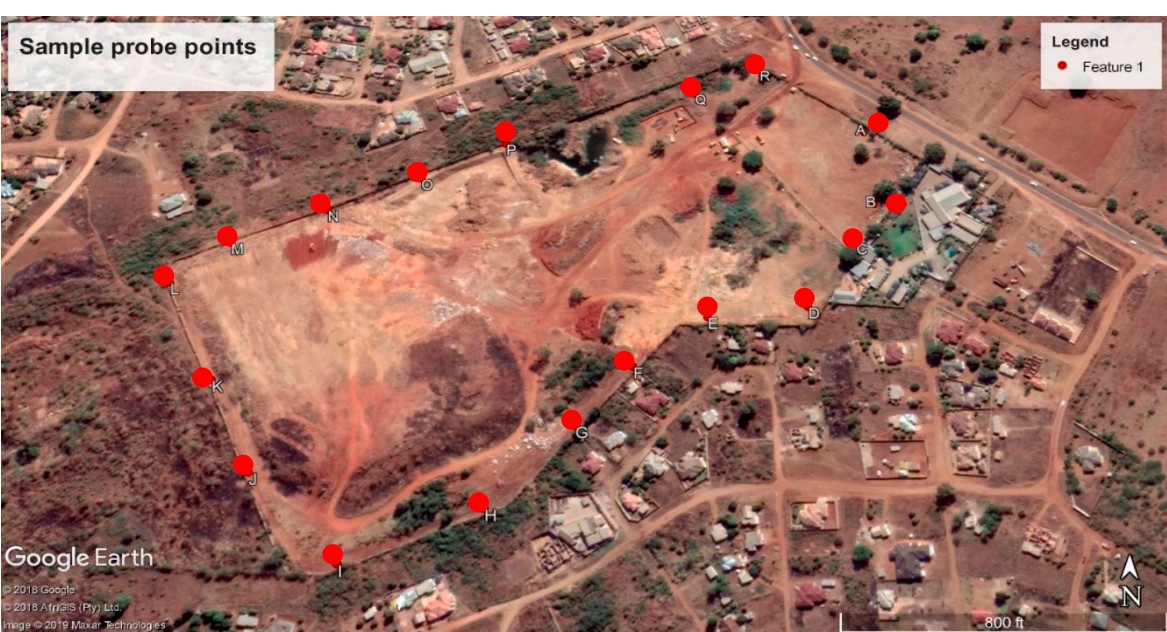

**Figure 3.** Aerial view of the landfill site showing the different points the LFG monitoring probes were installed at the boundary of the landfill. Note: Point A–R shows the points where the monitoring probes were installed in the landfill.

A unit monitoring probe consists of a cement top which was painted red for easy visibility. In addition, the cement top was used to protect the probes from theft or vandalism and even land fires. The threaded cap was used as a connection in which the gas samples were collected from the probes and measured using the GA 2000 landfill gas analyzer (Geotechnical Instrument; Keison Products, Chelmsford Essex, UK). The protective surface casing was designed with a tight cap which prevented water instruction into the probes. Water intrusion into the corners of the probes was also averted with the cemented base of the monitoring probe. Water intrusion into the probes was not a problem during sample collection.

*2.3. Data Collection*

Direct readings of gas concentrations of the principal components of LFG ($CH_4$ and $CO_2$) were taken using the GA 2000 landfill gas analyzer, a hand-held infrared gas analyzer developed by Geotec (Keison Products, Chelmsford Essex, UK). The hand-held GA 2000 landfill gas analyzer uses dual-wavelength infrared cells to measure $CH_4$ and $CO_2$ and a built-in electrochemical cell to measure $O_2$ [9]. The instrument was used for the analysis of gas samples taken from the sampling ports installed within the landfill. The outer perforated PVC pipes help for the allowance of the flow of the LFG into the probes. The inner perforated PVC pipes also allow for the gases in the probes and trap the gases. The methodology of this study was developed in accordance with methods prescribed by the South African Bureau of Standards (SABS) referred to in the Standard Act, 1982 (Act 30 of 1982) [33].

Fay et al. [7] showed that when many borehole wells of varied head space depths are sampled, the longest time to reach a steady-state measurement is approximately 2 min. Therefore, this study adopted a monitoring length of 3 min to allow for enough settling time. This resulted in a split in the monitoring time into three different operations (baseline, sampling, and purging), each of which was observed for three minutes, and samples were analyzed after every three minutes. The baseline procedure was when the GA 2000 landfill gas analyzer was switched on and the supply valve was switched to the atmosphere to check if the sensor was powered up. It also allowed enough time for the infrared (IR)

sensors to warm up and stabilize. The baseline sampling procedure ensured that no residual landfill gas remained in the chamber of the gas analyzer from previous measurement cycles.

Subsequently, during the sampling operations, the sampling valve from the GA 2000 landfill gas analyzer was connected to the extraction point of the landfill monitoring probe. The sampling valve outlet of the probes was linked to the GA 2000 landfill gas analyzer instrument, and the instrument then pulled the gas from the sampling point for approximately 60 s and the readings were recorded. Later, the measurements were repeated for 30 s consecutively three times. This was done until the readings from the GA 2000 landfill gas analyzer were stable. The results from the four measurements were then compared and averaged to give a final result. Then, the purge procedure was performed, whereby it was ensured that the LFG was removed from the instrument's chamber.

### 2.4. Data Analysis for the Influence of Meteorological Factors on $CH_4$ and $CO_2$ Emissions

$CH_4$ and $CO_2$ emissions were measured using the GA 2000 landfill gas analyzer. The data were collected and stored in Excel. Meteorological data were obtained from the South African Weather Service. To determine Pearson's correlation and statistical difference between the meteorological data and the LFG, a simple *t*-test analysis was conducted using the Excel tool. In order to investigate if there is a correlation between the meteorological parameters (rainfall, barometric pressure, wind speed, and temperature) and the LFG concentration, Pearson's correlation and *p* values (0.05 significant level) were calculated. To understand the causal effects of meteorological factors influencing the $CH_4$ and $CO_2$ emissions, a regression analysis was conducted using Excel.

## 3. Results and Discussion

### 3.1. Methane Generation from Thohoyandou Landfill

The handheld GA 2000 landfill gas analyzer was used to monitor $CH_4$ and $CO_2$ from an active landfill site for a period of 2 years, during which the monitoring continued for the different LFG monitoring probes. In 2020, $CH_4$ concentration was above the threshold limit for most months except for July (0.95% *v/v*), September (0.54% *v/v*), and October (0.65% *v/v*). In addition, in 2021, $CH_4$ concentration levels were above threshold limits except for in the months of March (0.26%), April (0.31%), and May (0.24%) (Figure 4). However, in the months of April and May 2020, monitoring was not conducted in the landfill due to the SARS-CoV-2 coronavirus pandemic and the national lockdown in South Africa. Monitoring commenced again in the late winter season (June, July, and August) of 2020; there was an increase in $CH_4$ concentration with values of 1.46% *v/v* in the month of June. In July there was a decline in $CH_4$ concentration, which was pegged at the threshold limits of 1% *v/v*; however, an increase in the $CH_4$ concentration was recorded in the month of August 2020. In the winter of 2020, it was observed that the average $CH_4$ concentration was lower than the average summer results. This result is consistent with the results of Monster et al. [34] and Aghdam et al. [28]. In a review of the literature, Monster et al. [34] showed that the lateral $CH_4$ movement from old landfills was not detected at all during summer due to increased oxidation of the soil.

In addition, from early September to October of 2020 (the $CH_4$ exhibited lower concentration), it was observed that there was a breakdown of the bulldozer for compressing and daily covering of waste. Therefore, there was no form of daily compression and covering of the waste during that period, which resulted in high penetration of oxygen and moisture content into the waste piles, thereby increasing the decomposition process in the landfill. The high presence of moisture encouraged bacterial growth and transported nutrients and bacteria to all areas of the landfill. This brought about the increase in pungent odor, vermin, dogs, high piles of waste, and high moisture content in the waste buried. In addition, the presence of oxygen in the waste piles could have brought about the reduction in $CH_4$ generation. Similarly, Li et al. [35] suggested that the lack of appropriate daily covering of waste in landfills can lead to an increase in the spread of disease vectors affecting human health, potential fire hazards, offensive odors, litter blown around the landfill environment,

aesthetic problems, vermin and insects, and lateral movement of LFG. Monster et al. [34] further suggested that when a very thin layer of cover materials is used during the daily covering, it may not be sufficient to enable the lateral migration of LFG but will increase the vertical movement of the gases into the environment. In addition, it is important that the choice of cover materials does not in itself create an environmental nuisance such as dust, litter, or odor. Notwithstanding, from this study, during the machine breakdown, a decline in $CH_4$ concentration was observed, and $CH_4$ concentration was recorded to be below the threshold limit. In November 2020, $CH_4$ subsurface migration gained momentum following the repairs of the equipment and the improved daily cover of the waste piles. Because of the daily compression and covering of the waste piles, there was low intrusion of oxygen and moisture content in the waste piles; this reduced the degradation process and reduced the vertical movement of the gases. The daily cover of the waste piles enhanced the horizontal flow of the $CH_4$. Keenan et al. [36] also suggested that due to sufficient and efficient daily covering of waste piles, there is low infiltration of moisture and oxygen content increasing the methanogenic activities in the landfill as this improves the lateral migration of the gases. As a result, the probes that were installed in the boundaries of the landfill experienced higher $CH_4$ concentration readings at that time.

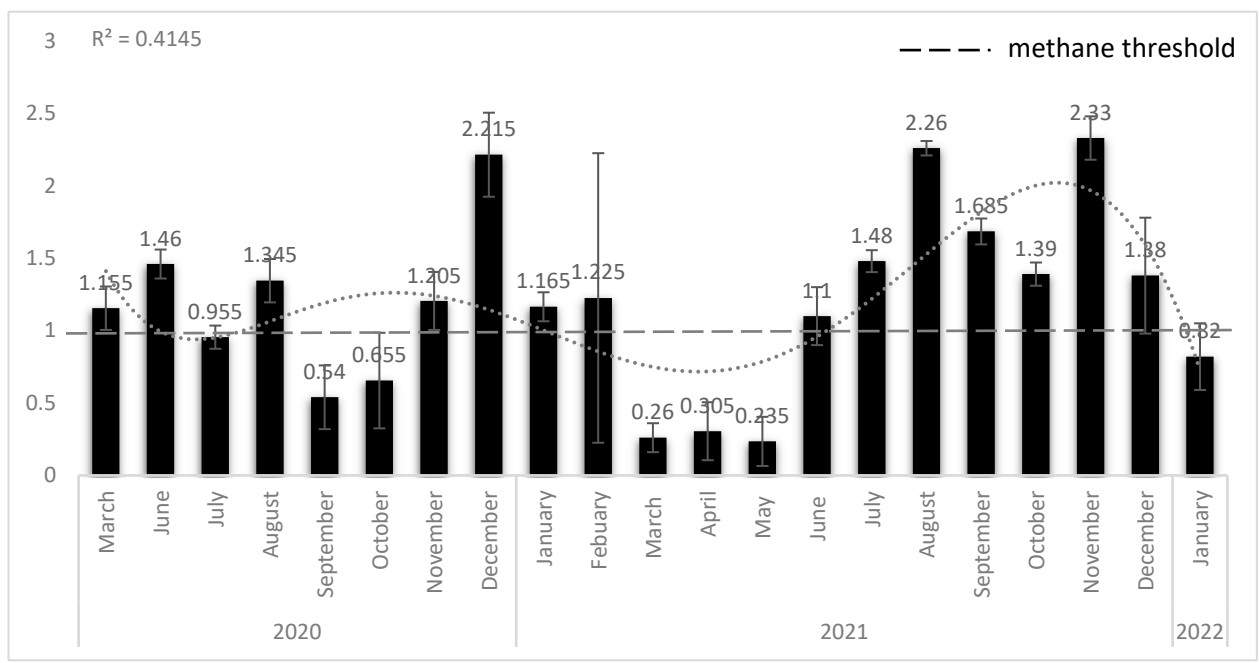

**Figure 4.** Average methane concentration observed during each month (March 2020–January 2022).

At the beginning of summer (December 2020, January and February 2021), there was a rapid spike in $CH_4$ concentration greatly exceeding the threshold limit. However, there was a subsequent decline in $CH_4$ concentration for the next three months (March, April, and May of 2021). Landfill activities and different meteorological conditions varied during this period (for example, introduction and siting of new cells, relocation of the leachate ponds, higher rainfall, and increase in temperature). Similarly, Park and Shin [13] showed that the LFG generation rate decreases from summer to winter in their study at Inchon city, Korea. According to Park and Shin [13], the decrease in surface efflux rate was very low because the surface pores were decreased by low temperatures and the compactness of solid ground during winter. Then during summer, the efflux rate of the LFG rapidly increased due to increased precipitation and high temperature. In India, Gollapalli and Kota [37] conducted a study on the $CH_4$ emission from an Indian landfill for a period of one year and obtained results similar to those of Park and Shin [13]. This study observed that $CH_4$ emissions were highest in summer and lowest in winter. This could be due to

higher temperatures in summer (30.5 °C) than in winter (19.7 °C). The diurnal variation in emissions indicated that the emissions follow a trend similar to temperature for all seasons. Gollapalli and Kota [37] also observed that moisture content in the landfill is very critical for the production of LFG. If adequate moisture is not available, then gas formation will not proceed and, in some cases, may not even start at all. In a study conducted in China, three-year monitoring of $CH_4$ and $CO_2$ effluxes at a large and well-managed final covered landfill, it was observed that $CH_4$ efflux in winter was higher than that in other seasons for most areas of the landfill [35]. Li et al. [35] attributed the results of the study to the gas permeability and the $CH_4$ oxidation capacity of the cover layer.

The months of March, April, and May 2021 exhibited the lowest $CH_4$ concentrations, below the threshold limits, following disturbances and increased activities in the landfill. Activities such as the construction of a new cell brought about deep excavations of the topsoil in the landfill. Therefore, this disturbed the flow of the $CH_4$ gas both laterally and vertically. This brought about the high introduction of oxygen and precipitation into the waste pile. Oxygen in landfill has to be used up first for the methanogenic bacteria to start producing $CH_4$. When the waste pile is not compacted properly when buried or frequently disturbed, more oxygen is introduced, so the oxygen-dependent bacteria live longer and produce $CO_2$ and water for longer periods. When the waste piles are very compacted, $CH_4$ generation will begin earlier as the aerobic bacteria are replaced by methane-producing anaerobic bacteria. $CH_4$ gas begins to be generated by the anaerobic bacteria only when the oxygen in the landfill is used up by the aerobic bacteria; therefore, more oxygen present in the landfill will slow methane production [8]. In addition, the majority of the gases could escape from the excavated topsoil of the landfill into the ambient air. This brought about the low readings from the monitoring probes.

In the winter (June, July, August) of 2021, the results showed a steady rise in $CH_4$ concentration from the month of June through August. The average results observed in the winter of 2021 were recorded to be almost the same as the results recorded during the previous summer season. There was no significant variation between the average results of the winter and summer seasons, as recorded in Figure 4. The rise in $CH_4$ concentration from landfills during winter is due to the decrease in temperature. Additionally, the decrease in temperature reduces the rate of $CH_4$ oxidation, which further contributes to the rise in $CH_4$ concentration [28]. Following the months of September through December of 2021 there were observed to be varying $CH_4$ concentrations above the threshold limit. This is a course of concern to landfill management, and measures must be put in place to reduce methane subsurface migration to avoid unexpected fire explosions at $CH_4$ concentrations greater than 5% in the air [18].

In summary, for the first year, 2020, during the data collection, it was observed that the $CH_4$ emission concentrations ranged from 0.54 to 2.215% *v/v*. The results further showed that the lowest concentrations of methane emissions were recorded for the months of September and October 2020, with values of 0.54 and 0.655% *v/v*, respectively. The month of December 2020 exhibited the highest amount of $CH_4$ emissions with a value of 2.215% *v/v*. Furthermore, for the year 2021, the months of March (0.26% *v/v*), April (0.31% *v/v*), and May (0.24 *v/v*) showed the lowest amount of methane emission. The months of August (2.26% *v/v*) and November (2.33% *v/v*) recorded the highest methane emission concentrations. It was recorded that the months of March, June, August, November, and December, with average values ranging from 1.16% *v/v* to 2.22% *v/v*, all surpassed the permissible limit (Figure 4).

### 3.2. Carbon Dioxide Generation from Thohoyandou Landfill

After the installation of the monitoring probes, in March 2020, the $CO_2$ concentration was approximately 6.37% *v/v*, which above the threshold limits for $CO_2$ emissions (Figure 5). However, no measurements were conducted between April and May 2020 due to the national lockdown in response to the SARS-CoV-2. In the winter season, the month of June showed a considerable decline in the $CO_2$ concentration. Consequently, there was an

increase in trends of $CO_2$ emissions from June to August 2020. The average $CO_2$ concentration in winter was 14.82% $v/v$, and the average $CO_2$ concentration in summer was 15.27% $v/v$; there was just a slight difference between the two seasons. Similarly, [35] Li et al. (2020), in a study conducted in a landfill in China, observed that the $CO_2$ efflux increased in spring and peaked in the summer of that year, and then the $CO_2$ concentration decreased to a minimum in late autumn or early winter. This was mainly a result of the seasonal change in landfill cover and soil gas permeability. Elmi et al. [38] showed that $CO_2$ concentration was greater in winter than in summer. This was a result of higher evaporative losses in summer, which resulted in less waste moisture content; this brought about a limiting factor in the formation of anaerobic conditions. There continued to be rapid fluctuations in the $CO_2$ concentration until the end of the monitoring activity, irrespective of the machinery breakdown that occurred in the month of September 2020. This breakdown affected the daily covering of the waste piles. However, October and November of 2020 showed a decrease in $CO_2$ concentration. The breakdown of the waste led to the increase in oxygen in the landfill and thus an increase in the oxygen-dependent bacteria decomposing the waste and the byproducts of $CO_2$ and water. This could have led to the observed increase in $CO_2$ concentration in the month of September 2020. During the summer season, there was varying $CO_2$ concentration. At the peak of summer, in the month of January 2021, the highest $CO_2$ level was recorded when compared to other months. During the winter months of 2021, the month of June experienced the lowest $CO_2$ concentration of 3.55% $v/v$ for the duration of the monitoring exercise. During this period, there was increased activity in the landfill, including heavy excavations, relocation of the leachate pond, and the construction of the new cell.

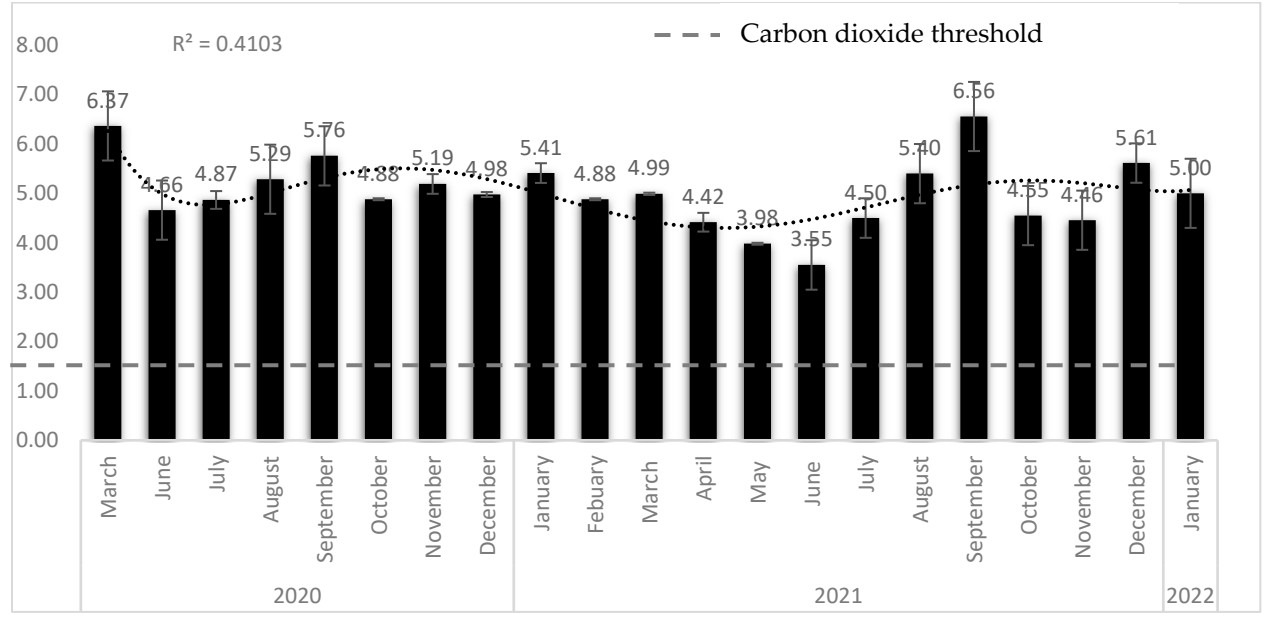

**Figure 5.** Average $CO_2$ concentration observed during each month (March 2020–January 2022).

The summary of the concentration data in Figure 5 shows that for all the monitoring months of this study, $CO_2$ emission was above the threshold level of 1.5% $v/v$ for the entire period of the monitoring process. This is a result of the decomposition of the high organic matter content in anaerobic conditions. When organic materials, such as food waste, are buried in landfills, they undergo microbial decomposition in the absence of oxygen, resulting in the production of $CO_2$ as a byproduct. Furthermore, the transportation of waste to and from landfills, as well as the equipment used to manage and cover the waste, also contributes to the high $CO_2$ concentration.

However, the $CO_2$ generation varied from time to time during the monitoring process. The minimum and maximum $CO_2$ concentrations obtained for the monitoring process

(March 2020 to January 2022) were 3.55% $v/v$ (June 2021) and 6.56% $v/v$ (September 2021), respectively. These variations were because of landfill disturbances, weather conditions, machine breakdown, and other underlying factors. Beirne et al. [9] showed that a partial blockage of an underground gas extraction pipeline restricted the volume of gas extraction and caused an increase in gas migration toward the sampling point. This blockage disturbed the subsurface flow of the gas component; meanwhile, once the blockage was removed, the $CO_2$ gas component fell below the threshold limit again.

### 3.3. Monitoring of $CH_4$ and $CO_2$ Concentrations from the Different Probes of the Thohoyandou Landfill

Table 1 shows the average concentrations of $CH_4$ and $CO_2$ emissions for the different monitoring probes. Over the monitoring period, it can be observed that the results obtained from the GA 2000 landfill gas analyzer for $CH_4$ concentration ranged from 0.26% $v/v$ to 2.56% $v/v$.

**Table 1.** Average $CH_4$ and $CO_2$ concentrations from the monitoring of the different probes in the landfill.

| Probes | Average $CH_4$ | Average $CO_2$ | Standard Deviation for $CH_4$ | Standard Deviation for $CO_2$ |
|--------|------|------|------|------|
| A | 0.56 | 5.14 | 0.31 | 1.3 |
| B | 0.30 | 4.58 | 0.17 | 1.31 |
| C | 0.48 | 3.71 | 1.63 | 1.79 |
| D | 0.26 | 4.56 | 0.21 | 0.5 |
| G | 1.58 | 3.86 | 1.05 | 1.14 |
| H | 0.69 | 3.59 | 2.28 | 1.78 |
| I | 2.30 | 6.47 | 0.2 | 0.81 |
| J | 2.33 | 6.65 | 0.3 | 1.1 |
| K | 1.60 | 6.44 | 0.85 | 2.35 |
| M | 2.56 | 5.96 | 1.77 | 1.37 |
| N | 0.55 | 5.92 | 0.17 | 1.19 |
| O | 0.55 | 6.68 | 0.45 | 1.44 |
| P | 0.40 | 6.21 | 0.23 | 1.38 |
| Q | 0.60 | 6.08 | 0.38 | 2.27 |
| R | 0.50 | 6.74 | 0.25 | 2.47 |

Zhang et al. [39] suggest that the lateral movement of the $CH_4$ concentration tends to diffuse as it travels below the earth's surface; also, some of the gases tend to escape from the surface of the soil if not covered properly, as is the case of Thohoyandou landfill. This can be applicable to the monitoring probes A and B, which were observed to have the lowest $CH_4$ concentrations with values of 0.56% $v/v$ and 0.30% $v/v$, respectively. Probes A and B were installed close to the entrance gate of the landfill and furthest away from the disposal site. Monitoring probes C and D were located very close to a hotel and had concentrations of 0.48% $v/v$ and 0.29% $v/v$, respectively. These concentrations were below the threshold limit of 1% $v/v$; however, it is pertinent to conduct constant monitoring of the hotel buildings because the concentration of $CH_4$ found in that area fluctuates and could be influenced by several meteorological factors (such as rainfall and high temperature or pressure) and physical activities conducted in the landfill. LFG can migrate from a landfill through the soil into outdoor air as well as the indoor air of nearby buildings. LFG in outdoor air can enter a building through doors, windows, and ventilation systems [40]. Scheutz and Kjeldsen [20] emphasized that one of the main reasons for monitoring LFG emissions is because of their health implications and the risk of off-site gas migration to buildings and structures. Monitoring probes E, F, and L were vandalized; therefore, we were not able to collect data from them.

Monitoring probes G and H were installed in one of the first cells constructed in the landfill; this cell had been covered for several years. As shown in Table 1, the average $CH_4$ concentrations for probes G and H were 1.58% $v/v$ and 0.69% $v/v$. The average concentration sampled from probe G exceeded the threshold limits; however, the levels

from probe H were somewhat closer to the threshold limit for $CH_4$ concentration. This shows that the decomposition process of buried waste is still ongoing even after many years. Thereby, the LFG continues to be generated; thus, constant monitoring in landfills needs to be carried out even after the closure of the landfills [41]. The highest levels of $CH_4$ were recorded in probes I (2.3% $v/v$), J (2.33% $v/v$), and M (2.56% $v/v$). These monitoring probes were situated closer to the current dumping site in the landfill. However, monitoring probe M was situated closer to an already closed cell in the landfill. Probes G, I, J, K, and M all exceeded the maximum limits for $CH_4$ emissions. This is approximately 33% of the total monitoring probes installed in the landfill.

The $CO_2$ concentration emission from the landfill ranged from 3.40 to 6.74% $v/v$. The monitoring probe H had the lowest emissions, with a value of 3.59% $v/v$ $\pm$ 1.78. However, it was observed the probes I (6.47% $v/v$), J (6.65% $v/v$), K (6.44% $v/v$), O (6.68% $v/v$), and R (6.74% $v/v$) recorded the highest concentrations from the landfill. Furthermore, it was observed that all the monitoring probes exceeded the emission threshold limits for carbon dioxide. Beirne et al. [9], in a study on an autonomous greenhouse gas measurement system for analysis of gas migration on an Irish landfill site, showed that $CH_4$ gas remained below the threshold limit of 1.0% $v/v$ throughout the experiment period. Similar to this current study, the recorded $CO_2$ concentration level varied over the duration of the data presented and also exceeded the threshold limit (1.5% $v/v$). Furthermore, Pehme et al. [18] conducted a study on the spatial distribution of LFG degradation in biocover using the GA 2000 landfill gas analyzer. It was observed that the highest value of $CO_2$ recorded was 1.0% $v/v$. The study concluded that $CO_2$ migration to the atmosphere was low and the gases fluctuated according to the seasons.

### 3.4. Average $CH_4$ and $CO_2$ Comparison

There was a weak correlation between $CH_4$ and $CO_2$ concentrations (R = 0.18) ($p < 0.001$). Higher levels of $CO_2$ were recorded during the course of the study (Figure 6). This could be because $CO_2$ is generated not only from the biodegradation process but also from the oxidation of $CH_4$ and from soil respiration. $CH_4$, once generated, can move through the cover soil and become oxidized into $CO_2$, leading to increased $CO_2$ emissions. Heavy excavations and diggings in the landfill can disturb the LFG generation and the flow of the gases. This activity will introduce oxygen into the landfill, thereby disturbing the anaerobic process and increasing the aerobic process in the landfill with $CO_2$ and water being the byproducts. Since the $CH_4$ generation (anaerobic process) is disturbed by the introduction of oxygen, there will be less $CH_4$ generation and subsequently lower emissions. This could be what leads to the higher concentration of $CO_2$ emission from the landfill. Similarly, Popița et al. [42] show that $CO_2$ emissions were observed to be higher than $CH_4$ emissions. Pinheiro et al. [43] explained that the oxidation of $CH_4$ within the landfill, the poor LFG collection system, and the pressure are the reasons the $CO_2$ concentration was higher than the $CH_4$ concentration. Similar results were found by Li et al. [35]. Alternatively, Pehme et al. [18] observed that at the highest LFG gaseous emissions, $CH_4$ emission was higher than $CO_2$ emission from a landfill. However, the trend in the concentration of $CH_4$ over time was an overall decrease.

Table 2 shows that during the winter season of 2020, $CH_4$ and $CO_2$ concentrations were correlated with a value of $-0.73$, which was statistically significant at $p < 0.001$. As shown in Figure 6, the $CO_2$ concentration showed a constant rise whereas the $CH_4$ concentration showed a constant decline over the period. This implies that as the $CH_4$ concentration increases, the $CO_2$ concentration reduces. During the winter period, with lower temperatures and lower rainfall, we would expect the soil respiration and a higher rate of $CH_4$ oxidation, especially in areas where there is a hotspot area (high LFG concentration detected). A similar result was observed by Xu et al. [44]: $CH_4$ and $CO_2$ emissions had a high linear correlation during the winter season. A different correlation was observed in the winter of 2021; Table 2 shows that $CH_4$ and $CO_2$ were correlated with a value of 0.64. As the $CH_4$ emission increased, the $CO_2$ emission also increased. $CO_2$ respiration and $CH_4$

oxidation are highly dependent on temperature, and other anthropogenic activities in the landfill can influence $CH_4$ and $CO_2$ emissions. According to Czepiel et al.'s [27] study, the oxidation rate in cover soil was 30% $CH_4$ generated inside the landfill in summer; however, in winter 0% $CH_4$ emission was reported. The diurnal variation in photosynthetic $CO_2$ uptake by leaves from vegetation grown on the cover soil or around the landfill might affect the LFG emissions, especially in the summer season, since leaf $CO_2$ uptake is driven by photosynthetically active radiation. This could also change the ratio of $CH_4$ and $CO_2$ emission rates.

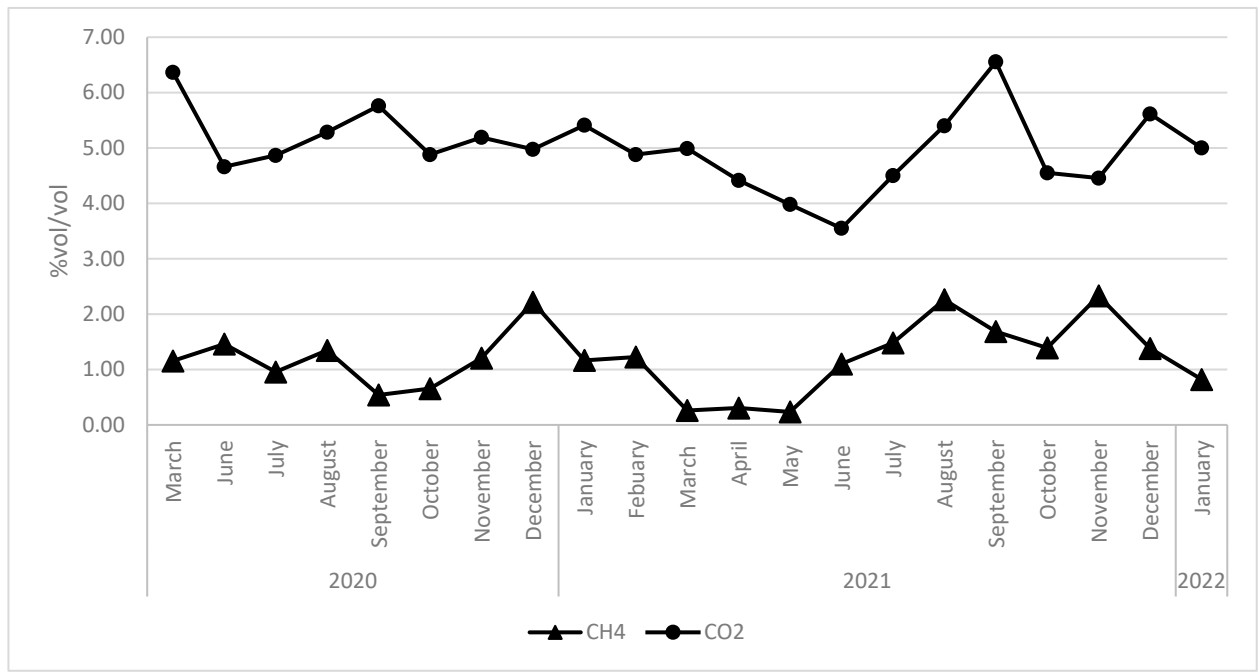

**Figure 6.** Average $CO_2$ and $CH_4$ concentrations observed during each month (March 2020–January 2022).

**Table 2.** Pearson's correlation coefficients and *p*-values between the $CH_4$ and $CO_2$ emissions during the different seasons.

|  | **Pearson Correlation** | ***p*-Value** |
| --- | --- | --- |
| Winter 2020 | −0.73 | $p < 0.001$ |
| Summer 2020 | −0.07 | $p < 0.001$ |
| Winter 2021 | 0.64 | $p < 0.001$ |
| Summer 2021 | 0.60 | $p < 0.04$ |

Table 3 shows that in the summer of 2020, Pearson's correlation between the $CH_4$ and $CO_2$ emissions was −0.07 (a weak correlation) and was statistically significant ($p < 0.001$). However, for the summer of 2021, Table 3 shows a Pearson's correlation between $CH_4$ and $CO_2$ at 0.6, which was statistically significant at $p < 0.04$. There was a strong relationship between the $CH_4$ and $CO_2$ concentrations. This could be because of higher soil $CO_2$ respiration and higher oxidation rate of $CH_4$ from the top cover soil. As a result, the $CO_2$ emission rate was probably higher than the $CO_2$ production rate associated with $CH_4$ production, while the $CH_4$ emission rate was lower than the production rate.

**Table 3.** Pearson's correlation coefficients and p-values between the selected meteorological parameters and LFG data during the periods studied in 2020 and 2022.

| Meteorology | $CH_4$ | | $CO_2$ | |
|---|---|---|---|---|
| | Pearson Correlation | *p*-Value | Pearson Correlation | *p*-Value |
| Temperature | 0.31 | $p < 0.001$ | 0.39 | $p < 0.001$ |
| Rainfall | −0.17 | $p < 0.01$ | 0.10 | $p < 0.001$ |
| Wind speed | 0.39 | $p < 0.001$ | 0.10 | $p < 0.001$ |
| Barometric pressure | −0.10 | $p < 0.001$ | −0.25 | $p < 0.001$ |
| $CO_2$ | 0.18 | $p < 0.001$ | | |

*3.5. Influence of Meteorological Conditions on the $CH_4$ and $CO_2$ Levels*

LFG generation is dependent on several meteorological factors such as temperature, wind speed, rainfall, barometric pressure, and humidity of the environment. This study looks into the impacts of ambient temperature, rainfall, barometric pressure, and wind speed on the $CH_4$ and $CO_2$ emissions from the Thohoyandou landfill. Firstly, Pearson's correlation was determined to understand the relationship between the meteorological data and the LFG emissions. Furthermore, a regression analysis was conducted to determine to what extent the meteorological conditions influence the LFG emissions (Tables 3–5).

**Table 4.** Regression analysis of $CH_4$ and meteorological data.

| Hypothesis | R Squared | Adjusted R Squared | Mean Square | F | *p*-Value |
|---|---|---|---|---|---|
| H1 | 0.098 | 0.051 | 0.39 | 2.067 | 0.17 |
| H2 | 0.029 | −0.023 | 0.10 | 0.559 | 0.46 |
| H3 | 0.15 | 0.11 | 1.122 | 3.36 | 0.082 |
| H4 | 0.011 | −0.042 | 0.079 | 0.203 | 0.66 |

**Table 5.** Regression analysis of $CO_2$ and meteorological data.

| Hypothesis | R Squared | Adjusted R Squared | Mean Square | F | *p*-Value |
|---|---|---|---|---|---|
| H1 | 0.15 | 0.11 | 1.52 | 3.37 | 0.82 |
| H2 | 0.01 | −0.042 | 0.10 | 0.20 | 0.66 |
| H3 | 0.091 | 0.043 | 0.92 | 1.90 | 0.18 |
| H4 | 0.059 | 0.009 | 0.59 | 1.18 | 0.29 |

Table 3 shows Pearson's correlation coefficients and the *p*-values between the selected meteorological parameters (ambient temperature, rainfall, barometric pressure, and wind speed) and the $CH_4$ and $CO_2$ concentrations.

Tables 4 and 5 show the regression analysis of the meteorological data and the LFG concentrations. The study hypothesizes that there is a significant influence of the meteorological data on the LFG concentration. H1 hypothesizes that there is a significant influence of barometric pressure on the LFG concentrations ($CH_4$ and $CO_2$) recorded. H2 hypothesizes that there is a significant influence of ambient temperature on the LFG concentrations ($CH_4$ and $CO_2$). H3 hypothesizes that there is a significant influence of rainfall on the LFG concentrations ($CH_4$ and $CO_2$). H4 hypothesizes that there is a significant influence of wind speed on the LFG concentrations ($CH_4$ and $CO_2$).

3.5.1. Barometric Pressure

The barometric pressure showed a weak correlation with $CH_4$ and $CO_2$ emission concentrations with values of −0.10 and −0.25, respectively, during the duration of the

study (Table 3). This means there is a slightly inverse relationship between the changes in $CH_4$ and $CO_2$ concentrations and barometric pressure. However, correlations of barometric pressure and the LFG concentration were observed to be statistically significant at $p < 0.001$. The *p*-value shows that there is a relationship between the changes in LFG concentrations and the barometric pressure.

Rachor et al. [45] showed a similar result: two $CH_4$ emission hotspots (1 and 20) showed an inverse relationship with pressure in a Pearson's correlation analysis (R = −0.64). The hotspot was distinguished by its unique location near the top of the landfill, where the cover is incredibly thin and exposure to the effects of wind, atmospheric pressure, and temperature is greatest. Similarly, Delgado et al. [46] showed that a strong inverse correlation emphasizes how pressure enhances air access into the landfill surface layers; this prevents $CH_4$ from escaping into the atmosphere. The authors further stated that the pressure changes had little effect on the $CH_4$ emissions fluxes, most likely because of the high degree of soil cover compaction in the landfill surface.

Figure 7 illustrates the relationship between the barometric pressure and $CH_4$ concentration; it was evident that the barometric pressure at its peak or lowest points resulted in decreasing $CH_4$ concentrations. For example, in September 2020, the $CH_4$ concentration recorded an all-time low of 0.54% *v/v*, and there was also a decreasing barometric pressure. The barometric pressure continued to decrease throughout the year; however, December 2020 showed the highest $CH_4$ concentration reading at 2.22% *v/v*. Furthermore, in 2021, the lowest $CH_4$ concentration was identified in the month of May at a value of 0.24% *v/v*, and the barometric pressure was observed to be at one of its highest values during that same period at a value of 95.19 kPa. The month of July 2021 recorded the highest barometric pressure of 95.68 kPa, and also the highest $CH_4$ concentration was observed the next month in August 2021 (2.26% *v/v*). These results do show some areas of an inverse relationship between the barometric pressure and $CH_4$ concentrations for the duration of the study.

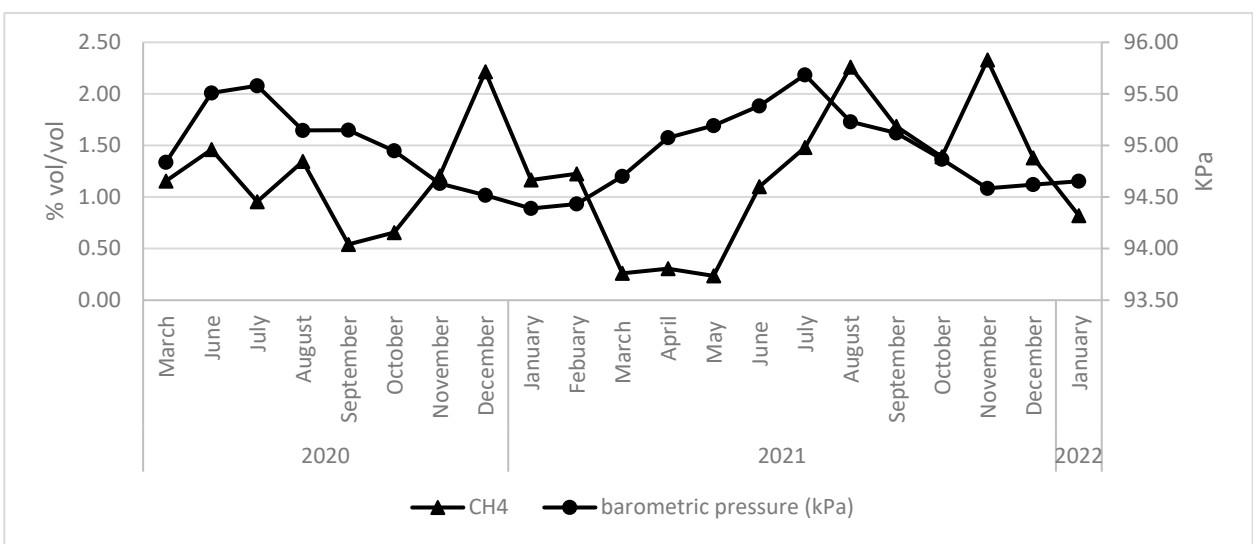

**Figure 7.** Average $CH_4$ concentration observed during each month (March 2020–January 2022) with barometric pressure.

The dependent variable ($CH_4$ concentration) was regressed on predicting variables (barometric pressure) to test the hypothesis H1 (Table 4). Table 4 shows that barometric pressure does not significantly predict landfill gas emissions ($p = 0.17$). However, there is a 9.8% chance that the changes in barometric pressure influence the $CH_4$ concentrations at $R^2 = 0.098$. That is, the changes in the increase in barometric pressure have a 9.8% chance of influencing the $CH_4$ concentration emitted from the landfill. Higher barometric pressure infuses more air into the landfill, which affects the stability of the LFG generation and overall concentration. When air is introduced into the landfill, the presence of oxygen is

increased, thereby increasing the aerobic bacteria activities in the landfill, producing more $CO_2$ and less $CH_4$ generation. During the study, there were different landfill activities that disturbed the daily operations of the landfill and eventually disturbed the LFG generation and emission in the landfill. Some disturbances in the landfill included the creation of a new cell, machine breakdowns, leachate pond relocation, closure of an old cell, construction of a fence around the landfill, and change in landfill management. Xu et al. [44] showed that landfill $CH_4$ emissions strongly depended on variations in barometric pressure; increasing barometric pressure suppressed the $CH_4$ emission, while decreasing barometric pressure improved the $CH_4$ emission, a phenomenon called barometric pumping. Similar results were also observed by Aghdam et al. [28] where a significant inverse relationship between $CH_4$ emissions and atmospheric pressure was observed with a linear regression ($R^2 = 0.95$).

Pearson's correlation ($-0.25$) shows a weak correlation between the barometric pressure and $CO_2$ concentrations. There is a significant difference between barometric pressure and $CO_2$ concentration ($p < 0.001$). This means that there is a relationship between barometric pressure and $CO_2$ emissions. In June 2020, the lowest concentration of $CO_2$ was recorded, at 4.66% $v/v$; however, there was sharp increase in barometric pressure at that time. All through the year 2020, there was a constant decline in barometric pressure, but the $CO_2$ concentration continued to range throughout the year. At the beginning of the year 2021, there was a constant increase in barometric pressure, as the increase continued, the records show a constant decline in $CO_2$ concentration reading (Figure 8).

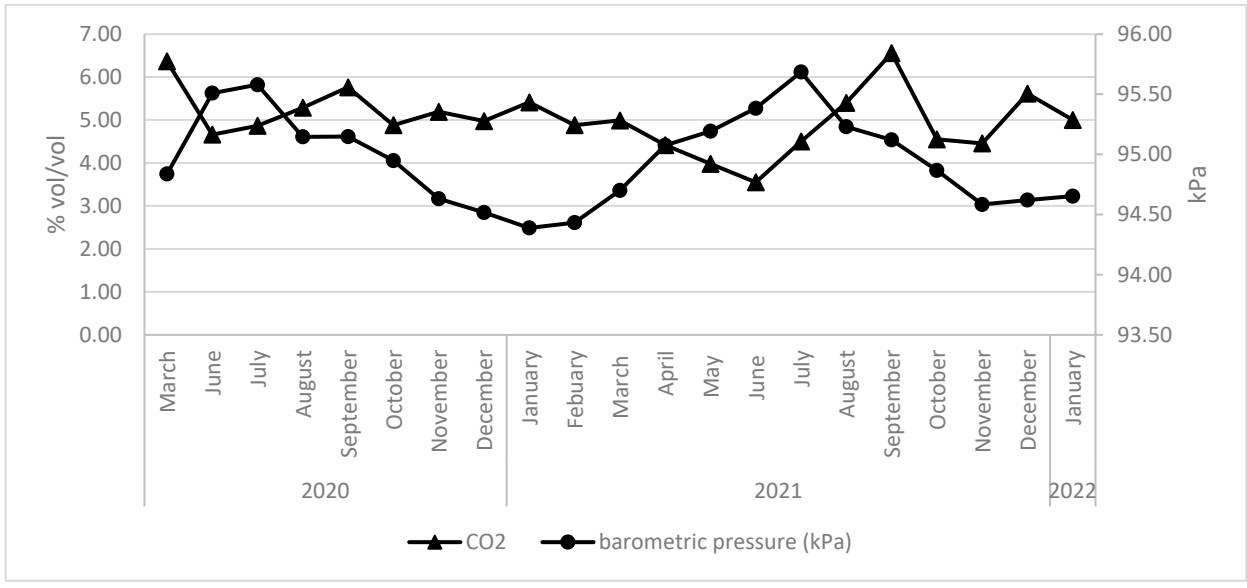

**Figure 8.** Average $CO_2$ concentration observed during each month (March 2020–January 2022) with barometric pressure.

The dependent variable ($CO_2$ concentration) was regressed on predicting variables (barometric pressure) to test the hypothesis H1 (Table 5). The barometric pressure does not significantly predict the $CO_2$ emission concentrations ($p = 0.82$). However, there is a 15% chance that the changes in barometric pressure influence the $CO_2$ concentrations ($R^2 = 0.15$) during the duration of the study. That is, the variations in the increase in barometric pressure have a 15% chance of influencing the $CO_2$ concentration emitted from the landfill.

3.5.2. Ambient Temperature

A weak correlation coefficient ($R = 0.31$) was observed between the ambient temperature and $CH_4$ emission and was statistically significant at $p < 0.001$ (Table 3). This means that there is a significant relationship between the ambient temperature and the $CH_4$ emissions from the landfill ($p < 0.001$). High temperature leads to higher microbial

activities in the landfill which in turn increase the $CH_4$ generation rates or $CH_4$ oxidation rates [28]. Aghdam [28] observed that $CH_4$ emissions were correlated with soil temperature. In 2021, the $CH_4$ concentration was at its lowest in May (0.24% $v/v$), and one of the lowest temperatures of the year (at 17 °C) was also recorded. The low temperature at that time influenced the $CH_4$ concentration in the landfill; low temperature reduces the activities of the bacteria in the landfill, thereby reducing $CH_4$ generation and emission [26]. In addition, it was observed that in November of 2021, the highest levels of $CH_4$ concentration (value at 2.33% $v/v$) and highest temperature levels (25.6 °C) were recorded (Figure 9).

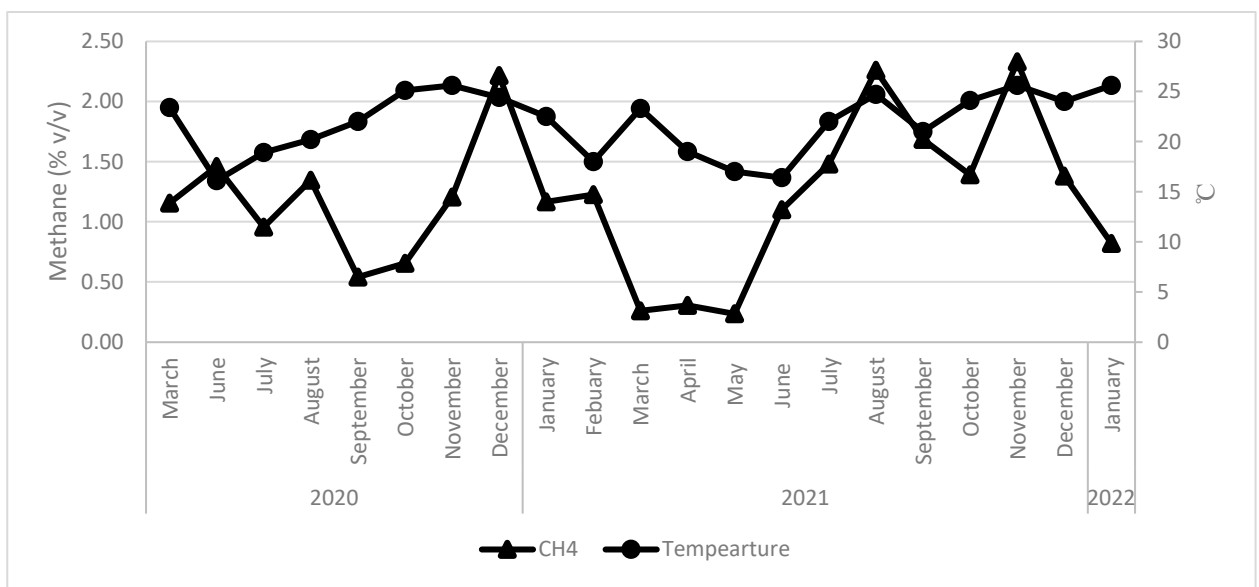

**Figure 9.** Average $CH_4$ concentration observed during each month (March 2020–January 2022) with ambient temperature.

$CH_4$ concentration was regressed on predicting variables (ambient temperature) to test the hypothesis H2 (Table 4). The results show that ambient temperature does not significantly predict the $CH_4$ emissions at $p$ = 0.46. However, there is a 2.9% chance that the changes in ambient temperature influence the $CH_4$ emissions ($R^2$ = 0.029). The variations in ambient temperature have a 2.9% chance of influencing the $CH_4$ concentration emitted from the landfill. Temperature changes have a far greater effect on LFG production in shallow landfills than in very deep landfills. This is because the bacteria are not as covered with cover materials as compared with very deep landfills where thick layers of soil cover the waste. Typically, warm temperatures increase bacterial activity, which in turn increases the rate of LFG generation. Meanwhile, colder temperatures inhibit bacterial activity [47,48]. The Thohoyandou landfill is a deep landfill, with some parts of the landfill covered with a thick layer of soil cover and some parts still receiving waste. These variations in the landfill cover material can lead to variations in temperature in the landfill. This can affect the overall LFG generation and emission from the landfill.

Table 3 shows a weak correlation between temperature and $CO_2$ concentration (R = 0.39). There is a relationship between temperature and $CO_2$ concentration ($p < 0.001$). The lowest $CO_2$ concentration was recorded in the month of June 2020 (4.66% $v/v$), and the lowest temperature reading was observed in the same month of June 2020 (16.1 °C) (Figure 10). In 2022, the lowest concentration of $CO_2$ was recorded in the month of June (3.55% $v/v$). Similarly, the lowest temperature was recorded in the month of June (value at 16.4 °C).

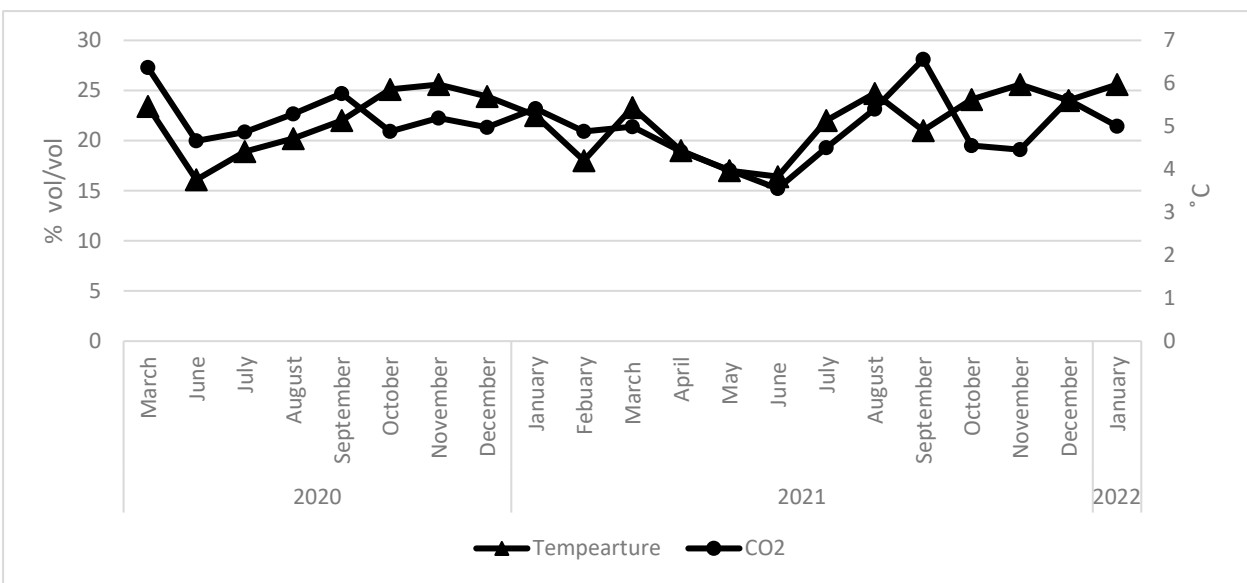

**Figure 10.** Average $CO_2$ concentration observed during each month (March 2020–January 2022) with ambient temperature.

The dependent variable ($CO_2$ concentration) was regressed on predicting variables (ambient temperature) to test the hypothesis H2 (Table 5). The results show that there is no significant difference between the ambient temperature and $CO_2$ emissions at $p = 0.66$. However, there is a 1% chance that the changes in ambient temperature influence the $CO_2$ concentrations at $R^2 = 0.01$. The changes in temperature has a 2.9% chance of influencing the $CO_2$ concentration emitted from the landfill.

In a comparison between the results shown in Figures 9 and 10, it was observed that $CH_4$ concentration seems to have more variation across the period compared to $CO_2$. For example, $CH_4$ varied between ~0.2 and 2.2% or 11 times and $CO_2$ varies between ~15 and 28% or less than 2 times. This could be a result of $CH_4$ oxidation in the cover material of the landfill. $CH_4$ oxidation is the conversion of $CH_4$ to $CO_2$ as the $CH_4$ gases move toward the cover material of the landfill. Some other factors that could influence the variations could be bacteria, temperature, wind speed, and rainfall. Additionally, the presence of other organic compounds in the landfill, such as volatile organic compounds (VOCs), can also contribute to the high variance in $CH_4$ over $CO_2$ [49]. VOCs can be broken down by bacteria in the landfill, releasing $CH_4$ and other gases as byproducts and reducing $CO_2$ concentrations.

3.5.3. Rainfall

Rainfall showed a weak correlation with $CH_4$ concentrations with a value of $-0.17$ for the duration of the study (Table 3). In addition, the relationship is statistically significant ($p < 0.01$). This means that there is a relationship between rainfall and $CH_4$ concentration. Rainfall increases the soil moisture content and decreases oxygen levels, which regulates nitrification and denitrification and also limits bacteria activities in the soil. Bian et al. [25] observed that the degree of decrease in $CH_4$ emissions was closely correlated with the rainfall intensity. This is because a more significant increase in water content under heavier rainfall leads to a great reduction in LFG movement due to reduced microbial $CH_4$ oxidation activity.

Figure 11 illustrates the relationship between rainfall and the $CH_4$ concentration. It shows that the lowest $CH_4$ concentration (0.54% $v/v$) occurred in the year 2020; low rainfall was recorded at that time (value at 1.2 mm), in September 2020. The highest $CH_4$ concentration was recorded in the month of December 2020 (value at 2.2% $v/v$), and there was an increasing amount of rainfall. This affirms that the increasing rainfall increased the moisture content in the topsoil and waste pile, thereby increasing bacteria activities and

$CH_4$ generation and emission. Due to the low rainfall intensity, the moisture content of the top cover soil was low and permeability was not obstructed; therefore, the top cover soil did not impede $CH_4$ emissions. For 2021, Figure 11 shows that $CH_4$ in its lowest concentration was observed in the month of May (0.24% $v/v$) and there was no rainfall recorded in that month. The low rainfall affected the moisture content in the landfill, reducing bacteria activities and thereby reducing $CH_4$ generation and emission. In addition, it was observed that in November of 2021, the $CH_4$ concentration exhibited the highest readings of $CH_4$ emissions from the landfill.

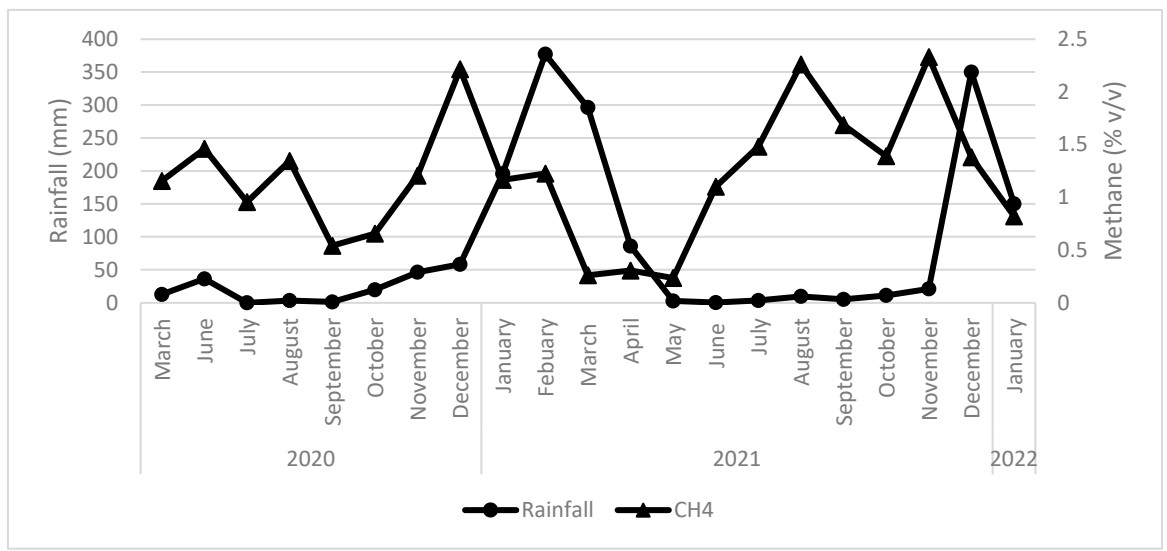

**Figure 11.** Average $CH_4$ concentration observed during each month (March 2020–January 2022) with rainfall.

The dependent variable ($CH_4$ concentration) was regressed on predicting variables (rainfall) to test the hypothesis H3 (Table 4). The rainfall does not significantly predict the $CH_4$ emissions at $p$ = 0.082. However, there is a 1.5% chance that the changes in rainfall influence the $CH_4$ concentrations at $R^2$ = 0.015 during the duration of the study. According to Yang et al. (2016) [50], heavy rainfall can cause the top cover soil of the landfill to be waterlogged, thereby decreasing the permeability of the top cover soil. Since LFG moves better in areas of high soil permeability, the decreased pores in the landfill cover restrict LFG emissions at that period of time. Notwithstanding, a certain amount of moisture content in the landfill can increase bacteria activities and transports nutrients, thereby increasing LFG concentration in an anaerobic condition. A moisture content of 40% or higher, based on the wet weight of waste, encourages maximum LFG generation, especially in a closed landfill. A waterlogged top cover soil will inhibit LFG emission [51].

The dependent variable ($CO_2$ concentration) was regressed on predicting variables (rainfall) to test the hypothesis H3 (Table 5). The results show that there is no significant difference between the rainfall and $CO_2$ concentrations emitted at $p$ = 0.18. However, there is a 9.1% chance that the changes in rainfall influence the $CO_2$ concentrations at $R^2$ = 0.091 during the duration of the study. Figure 12 gives the average $CO_2$ concentration observed during each month (March 2020–January 2022) with rainfall.

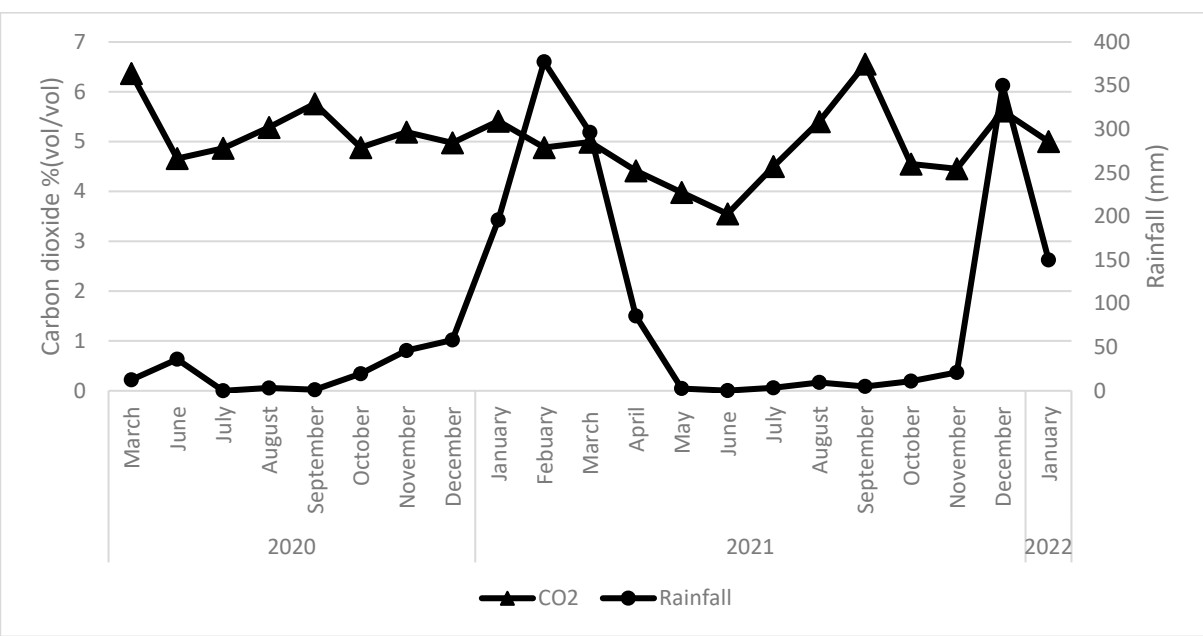

**Figure 12.** Average $CO_2$ concentration observed during each month (March 2020–January 2022) with rainfall.

3.5.4. Wind Speed

Table 3 shows a weak correlation between the wind speed and the $CH_4$ concentration for the duration of the study (value of 0.39), and the correlation is statistically significant ($p < 0.001$). Similar results were observed by Kissas et al. [52], who showed a weak correlation between $CH_4$ fluxes and wind speed (r = 0.21, $p < 0.001$). Figure 13 shows that the lowest $CH_4$ concentration was recorded in September 2020 at 0.54% $v/v$, and a high wind speed was recorded in the same month (value at 2.72 m/s). In addition, the month of October recorded the highest wind speed at a value of 2.84 m/s, but there was a slight increase in $CH_4$ concentration (value at 0.66% $v/v$). Furthermore, the lowest concentration of $CH_4$ in 2021 was observed in the month of May at 0.24% $v/v$ and with an average wind speed of 1.81 m/s. In addition, the highest concentration of $CH_4$ in 2021 was observed in the month of November at 2.33% $v/v$ and an average wind speed of 2.76 m/s. These observations did not have any correlation. Several studies have identified that wind induces advection as one of the dominant $CH_4$ emission mechanisms in windy conditions at landfills [28,53]. Wind blowing across a landfill can cause a pressure difference, which is the driving force for advective gas transportation. This means that a strong wind speed can create a pressure difference between the landfill body and the landfill surface, which in turn can affect LFG generation and emissions from landfills.

$CH_4$ concentrations were regressed on predicting variables (wind speed) to test the hypothesis H4 (Table 4). Wind speed does not significantly predict the $CH_4$ at $p = 0.66$. However, there is a 1.1% chance that the changes in wind speed influence the $CH_4$ concentrations at $R^2 = 0.011$ during the duration of the study.

The predominant perception is that the effect of wind speed on LFG emissions is indirect, as it creates the pressure-pumping effect that facilitates advective gas transportation. Moreover, Kissas et al. [52] suggested through model simulations how increased wind speed brings about changes in the pressure gradient between the soil and the atmosphere, resulting in increased surface $CH_4$ emissions.

Table 3 shows a weak correlation between wind speed and $CO_2$ concentration with a value of 0.10, which was very statistically significant ($p < 0.001$). Figure 14 shows the results observed in the correlation between wind speed and $CO_2$ concentrations. The lowest $CO_2$ concentration was recorded in the month of June 2020 (value at 4.66% $v/v$); also, in the month of June 2020, the wind speed was recorded at one of its lowest values (1.91 m/s).

$CO_2$ concentration was regressed on predicting variables (wind speed) to test the hypothesis H4 (Table 5). The results show that there is no significant difference between wind speed and $CO_2$ concentrations emitted $p = 0.029$. However, there is a 5.9% chance that the changes in rainfall influence the $CO_2$ concentrations at $R^2 = 0.059$ during the duration of the study.

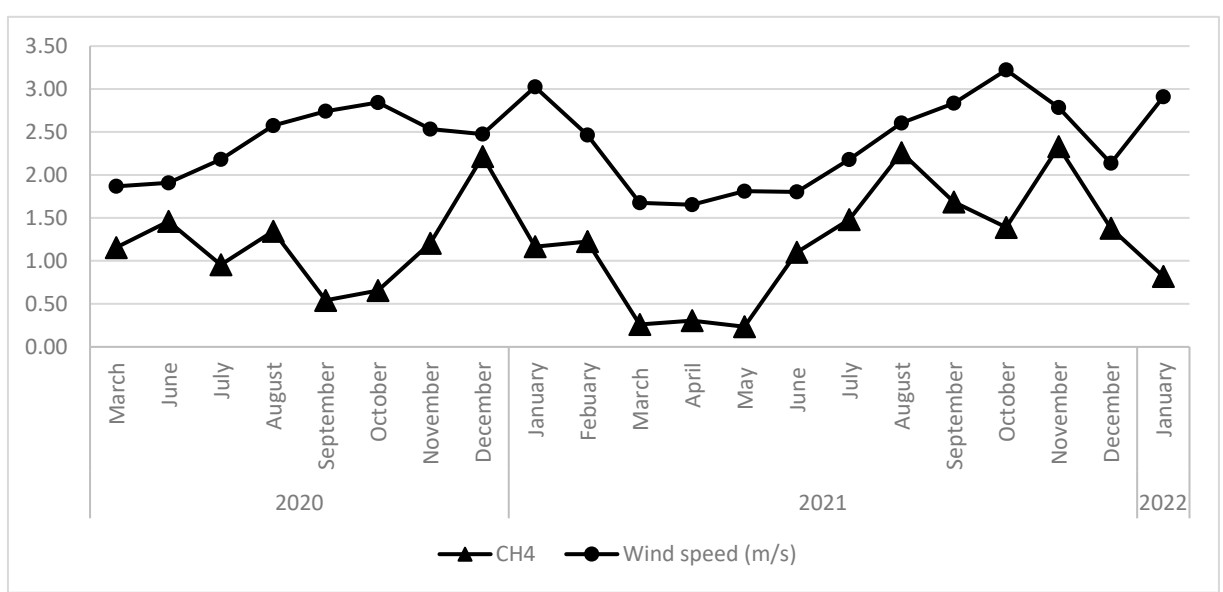

**Figure 13.** Average $CH_4$ concentration observed during each month (March 2020–January 2022) with wind speed.

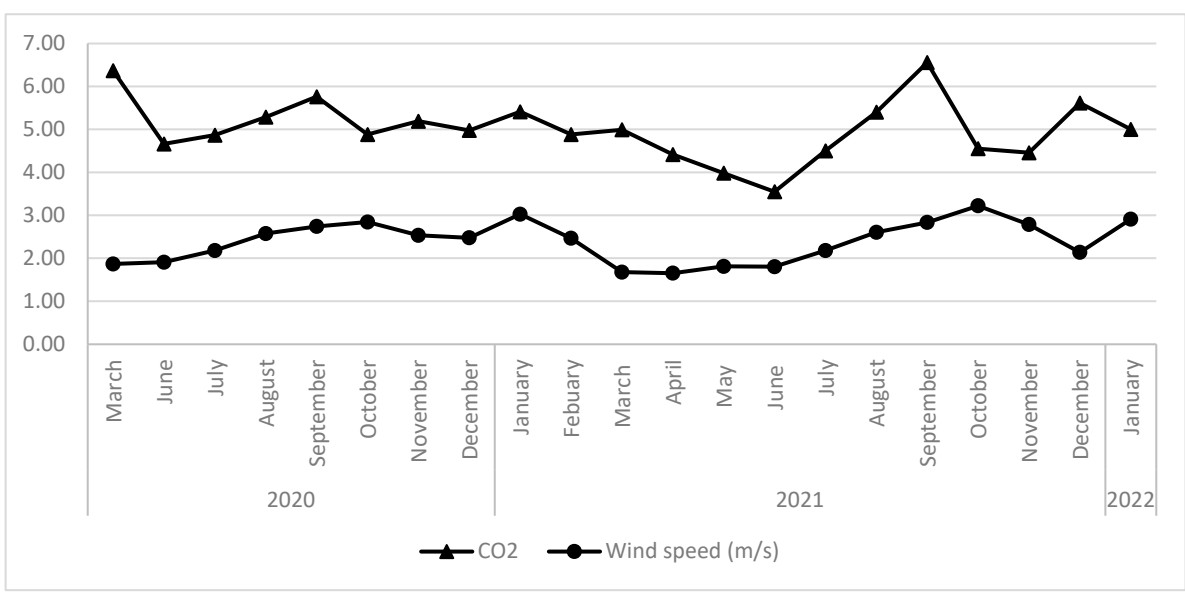

**Figure 14.** Average $CO_2$ concentration observed during each month (March 2020–January 2022) with wind speed.

## 4. Conclusions

Landfill gas monitoring probes were deployed on a landfill site, and LFG monitoring for a duration of two years (November 2019–January 2022) was successfully conducted. This study validates that LFG migrates along the subsurface of the landfill through the pores and cracks of the soil. The study concludes that $CH_4$ concentrations were observed to be lowest in areas far away from the landfill activities such as the entrance of the landfill.

Higher concentrations of $CH_4$ were observed in areas where the landfill cells have been closed for a long time. However, the highest concentrations of $CH_4$ were recorded in areas closer to the current dumping of waste. These high $CH_4$ concentrations were above the South African $CH_4$ emission limits from landfills. $CO_2$ concentrations for the duration of the study surpassed the South African $CO_2$ emission limits from landfills.

Furthermore, the study concludes that the $CH_4$ and $CO_2$ generation and emission are complex processes influenced by landfill activities and meteorological conditions around the landfill. All meteorological conditions that were selected for this study, namely barometric pressure, temperature, rainfall, and wind speed, were either negatively or positively correlated with the LFG concentration and were all statistically significant.

The study relates to sustainability and demonstrates the importance of monitoring LFG emissions in order to ensure that they are within the South African emission limits. This is important for maintaining a sustainable environment, as LFG emissions can have a negative impact on air quality and climate change. By monitoring these emissions, we ensure that they are kept within acceptable limits, and we can help to protect the environment and promote sustainability.

Furthermore, this information can be used to inform policymakers and regulators about the need to set appropriate limits for landfill gas emissions and to ensure that landfill sites are operating within those limits.

**Author Contributions:** Resources, S.P.; Writing—original draft, P.O.N.; Supervision, R.M. and J.N.E. All authors have read and agreed to the published version of the manuscript.

**Funding:** This research was funded by Eskom Power Plant Engineering Institute (EPPEI) (grant number [E349]).

**Informed Consent Statement:** Not applicable.

**Data Availability Statement:** Not applicable.

**Acknowledgments:** The authors would like to thank Joe Malahlela from the University of Northwest for his relentless effort in making sure the monitoring probes were constructed and installed in the landfill.

**Conflicts of Interest:** The authors declare no conflict of interest.

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
