# Peer review of "Monitoring of Subsurface Emissions and the Influence of Meteorological Factors on Landfill Gas Emissions: A Case Study of a South African Landfill"

_sustainability, doi:10.3390/su15075989_

Round 1
Reviewer 1 Report
Dear Authors,
The reviewer has gone through the manuscript and has some queries as highlighted below:
1. Please summarize some of the major findings in the study using numerical quantification.
2. Literature review of the subject is weak. Please improve the quality of the literature presented in the Introduction section and use global references for the same. Some references [https://doi.org/10.1061/(ASCE)EE.1943-7870.0001647; https://doi.org/10.1061/(ASCE)HZ.2153-5515.0000442] may be included
3. Please highlight the specific objectives of the study.
4. Figure 3, why the different LFG monitoring probes were located at the side of the landfill site?
5. What is the percentage of organic waste amongst the total waste dumped at the study landfill site?
6. What precautions were ensured for the study during rainy seasons to prevent contamination of the samples?
7. please present the hypothesis testing results used in the study.
8. September -October 2020 was during COVID times - how was sampling carried out during this period?
9. Present the discussion on the seasonal variation of LFG emissions over the study period.
10. Explain the change in the behavioral patterns of methane and carbon dioxide generated over the study period.
11. Table 4 - explain the reason for the change in behaviour between CO2 and CH4 for the period 2020 and 2021
Author Response
Thank you for your contribution was very helpful in the construction of the manuscript
please see attachment

Reviewer 2 Report
-Page 1 Line 36-38 Introduction “it will be more impact if authors mention about SDG or COP26 that UN set up about zero emission policy, please see some reference at http://ojs.kmutnb.ac.th/index.php/ijst/article/view/5842/pdf_364”
-Page 3 Line 114-116 Therefore, it is important “the way to explain objective here is more like that this work is the investigation research. It will be more beneficial if authors could provide any studies cases. after the results from investigation research were done and the report is available , is there any case that these results were implemented to make change in LFG management?”
-Page 4 Fig 2 “how this design of probe prevent moisture accumulation? if moisture interfere gas analysis?”
-Page 6 Line 243-244 Due to the high presence “is there any supportive evidence of this work, or else other references?”
-Page 6 Line 258-259 However, during that period “how authors know that it's the vertical movement for gas mass?”
-Page 7 Line 311 average results “which result? please be specific”
-Page 7 Line 315 reduce methane “what's the concentration of ch4 accumulation to activate fire explosion?”
-Page 10 Line 389 y several meteorological “please give some examples”
-Page 13 Line 496 shows a weak negative correlation “it's a weak correlation or no correlation? what's the criteria”
-Page 14 Line 520 A weak positive “again, what's the criteria to determine if it's a weak or no correlation?”
-Page 14 Line 540-542 This is because the bacteria “bacteria are not insulated? what does it mean, please clarify”
-Page 15 Fig 8 “comparing between fig 8 and 9 results, it can be seen that CH4 concentration seems to have more variation across the period compared to CO2. For example, CH4 varied between ~ 0.2 - 2.2 or 11 time. for CO2 varies between ~15-28% or less than 2 time. Please explain”
-Page 16 Line 578-580 Figure 10 “in above paragraph, authors suggested that rainfall has weak correlation to CH4. But this result suggest the correlation. So what's the authors's opinion? it's confusing”
-Page 17 Line 615 Wind speed “from this manuscript , several factors analyzed here all show weak correlation. So authors please suggest which factor should be strong correlation. Authors may refer to other studies”
-Page 19 Line 661 There was a positive “based on the principle that CH4 and CO2 are generated in opposite condition, anaerobic and aerobic, respectively. So how's the correlation of these gases are weak”
Author Response

(The authors gave the same response as above.)

Reviewer 3 Report
Since the article is submitted to the Sustainability journal, it should show some connections to the sustainability issues. The authors of this article mostly described the results of landfill gas monitoring. The conclusive part doesn't provide any statement if the results of this study confirm the results of other studies or differ from them. Also, it's not clear how the results could be used for landfill management and how these results contribute to sustainability.
Article has some issues with text formatting. At lines 15,16,20 numbers 2 and 4 should be subscript. Figures and tables should be aligned with the text.
Author Response

(The authors gave the same response as above.)

Reviewer 4 Report
Introduction should be improvedand with more relevant literature on previous efforts to measure CO2 and CH4 in landfills. Also, establish at the end of introduction what the reader will see in the following sections.
Discussion should include practical and theoretical implications.
Methodology should show where the probes where located and why are the locations validated (by which method the probes where set?).
Abstract should be improved following the previous recommendations.
Author Response

(The authors gave the same response as above.)

Reviewer 5 Report
Abstract:
1.There is need to improve the result with the LFG migration process in the subsurface landfill where the title of this manuscript is “monitoring of subsurface migration”
Introduction:
1. Need to improve the recent references about LFG migration in the landfill and how the relationship between meteorological data with gas emission or gas production
Method:
1. How to analyze the LFG migration processes which the monitoring probe only places at the surrounding of the landfill, where is the control function or as initial condition?
2. How to monitor the migration of the LFG?
Result and discussion:
I can’t find the discussion of the migration process in subsurface of the landfill how it happens/pattern? and how the relationship with the meteorological data?
Conclusion:
How you can conclude that LFG migrates along the subsurface of the landfill 714 through the pores and cracks of the soil? Did this study analyze that and supported with the data monitoring?

Author Response

(The authors gave the same response as above.)

Reviewer 6 Report
-In the abstract, the background information section should rewrite and revised.
-What landfill gases should be monitoring is missing
-Pls. mention the season considered for this study
-It’s important to mention landfill age from where the sample was monitored
-Only two gases CH4 and CO2 level were monitored?
-Does the study follow any monitoring guidelines?
-Keywords should be revised
-Line no. 53-54; disconnected from the previous section, need hints before starting this sentence. Pls. introduce to the reader what is sub-surface and migration.
-At the beginning of the introduction section, should focus on why the study is conducted and what’s the major problem and its importance.
-the authors are requested to mention the season when the fire outbreak risk is high and low.
-the authors are requested to add some information about landfill ages and stages. Also should mention whether there have any national, or international monitoring guidelines that need to follow.
-add the objectives of the current study
-Figure 1; Study area map, should reproduce. Couldn’t obtain enough information.
-when reconnaissance survey was conducted?
-How did the authors select the sampling points?
-“GA 2000 landfill gas analyzer”, pls. write in a standard format.
-Section 3.1.: the authors are requested to explain the result of why CH4 concentration is high and low in particular months.
-Section 3.1. Methane generation from the Thohoyandou landfill; should be concise
-Line no 308-315; why rise in CH4 in the winter season? Need to explain.
-Section 3.2: CO2 generation; this section needs to be concise. And explain the reason CO2 emission is above the threshold limit.
-I think section 3.4; Average CH4 and CO2 emission comparison would come up before the section “Influence of Meteorological condition”.
-3.4. Influence of meteorological conditions on the CH4 and CO2 levels; the section should be concise.
-the conclusion needs to be more concise and have key findings.
-the references are relevant to the study
-few references need to update; the author is requested to cite some recently published articles from top journals.

Author Response
thank you
please see attached

Round 2
Reviewer 1 Report
Dear Authors,
The reviewer has gone through the manuscript and observed that the authors have incorporated the majority of the changes suggested by the reviewer, hence the manuscript is approved for publication.
Reviewer 2 Report
all comments were responsed, so it is recommend for publication
Reviewer 3 Report
Quality of Structure and Clarity are significantly improved compare to the previous version. Sustainability connections are clear now.
Reviewer 4 Report
Good job however I could see an improvement in the discussion with more theoretical implications